# Truthfulness in LLMs: A Layer-Wise Comparative Analysis of Representation Engineering and Contrast-Consistent Search

## Abstract

The rapid advancement of Large Language Models (LLMs) has intensified the need for greater transparency in their internal representations. This study presents a **layer-wise analysis of truthfulness storage in LLMs**, comparing two state-of-the-art knowledge probing methodologies: **Representation Engineering (RepE)** and **Contrast-Consistent Search (CCS)**. Our goal is to isolate truthfulness, defined as the factual accuracy of LLM outputs, from general knowledge encoded across model layers and to examine where and how this information is stored. RepE applies low-rank transformations within the model's internal vector space, while CCS leverages pre-trained fixed vectors with an additional transformation layer to define truthfulness. Through experiments on Google's Gemma models, evaluated across five diverse datasets, we find that truthfulness is embedded within pre-trained LLMs and can be amplified by specific input words. Our analysis reveals general trends in truthfulness storage and transferability, with CCS demonstrating greater stability in assessing truthfulness, while RepE exhibits potential in deeper layers but requires further refinement. Surprisingly, the truthfulness differences in the final layer, often considered the most critical, were statistically insignificant. This study provides empirical insights into the internal encoding of truthfulness in LLMs, highlighting the strengths and limitations of representation-based transparency methods.

## 1 Introduction

Large Language Models (LLMs) have gained widespread attention following the release of ChatGPT by OpenAI[1]. These models, characterized by their large-scale parameters and extensive pre-training datasets (Raffel et al., 2020; Achiam et al., 2023), significantly outperform traditional language models. Unlike task-specific supervised models, LLMs learn generalized representations through self-supervised pre-training, allowing them to adapt to a wide range of natural language processing (NLP) tasks (Devlin et al., 2018).This paradigm shift has transformed how computers process and generate human language. Beyond basic language comprehension, LLMs now demonstrate capabilities in information retrieval, conversational interactions, and creative text generation (Naveed et al., 2023). Their applications extend to law (document analysis), medicine (diagnostic support) and education (personalized tutoring) (Kaddour et al., 2023; Thirunavukarasu et al., 2023; Kasneci et al., 2023), highlighting their broad impact in different domains.

However, the increasing complexity and scale of LLMs introduce opacity in their internal mechanisms, leading to a"black-box" behavior that challenges interpretability. This lack of transparency raises concerns such as hallucinations, model misalignment, and unpredictable outputs (Kaddour et al., 2023; Hendrycks et al., 2021). Addressing these issues is critical for improving trust and reliability in AI-driven applications. Improving transparency could reveal latent capabilities, mitigate safety risks, and enable a more responsible deployment (Huang et al., 2024).

---

[1]GPT-3 was released for public use in 2022, though the first GPT model was introduced in 2018.

**Research Focus and Contributions** This study aims to advance transparency in LLMs by investigating how truthfulness, defined as "the factual accuracy of model output", is represented within model layers. Specifically, we compare two knowledge probing methodologies:

- Representation Engineering (RepE), which transforms the LLM's latent vector space to isolate truthfulness-related representations.
- Contrast-Consistent Search (CCS), which analyzes contrast patterns in the pre-trained representations of the model, using fixed vector structures with an additional transformation layer.

We conduct experiments on Google's latest Gemma models, validating findings across five diverse datasets to examine:

1. **Comparative Evaluation of Truthfulness Probing Methods:** We empirically compare RepE and CCS in terms of effectiveness, stability, and generalizability in identifying truthfulness within LLM representations.
2. **Layer-Wise Analysis of Truthfulness Encoding:** We investigate where and how truthfulness is stored across different layers of LLMs, identifying general trends and transferability patterns.
3. **Guidelines for Enhancing LLM Interpretability:** By distinguishing model-wide truthfulness patterns from method-specific effects, we provide insights to inform future research on LLM transparency, safety, and reliability.

These contributions provide a foundational step toward improving interpretability in LLMs, offering insights into how knowledge representations can be refined for more responsible AI deployment.

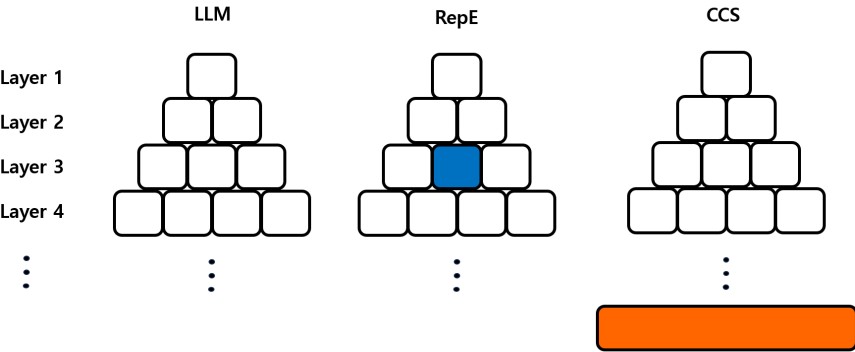

Figure 1: Conceptual framework comparing RepE and CCS for probing truthfulness in LLMs. CCS (right) extracts representations from the entire model and applies an external transformation via an additional function (orange block). In contrast, RepE (middle) modifies the LLM's internal representations by introducing additional vectors directly within specific layers (blue block). The baseline LLM architecture (left) illustrates the unmodified representation flow, providing a point of comparison.

## 2 LITERATURE REVIEW

### 2.1 INTERPRETABILITY IN LLMS

Efforts in explainable AI (XAI) have aimed to improve the transparency of LLMs by elucidating their internal processes and decision-making (Cambria et al., 2024). Techniques like saliency maps highlight influential input regions (Alqaraawi et al., 2020), while feature visualization identifies neuron activations (Nguyen et al., 2019). Post-hoc methods, such as SHAP and LIME, explain individual predictions (Kästner & Crook, 2023). Although these approaches offer valuable insights, they

fall short of fully explaining how internal representations encode concepts, leaving significant gaps in understanding model behavior at scale (Minh et al., 2022). Recent advances in LLM interpretability draw inspiration from cognitive neuroscience (Vilas et al., 2024), incorporating Sherringtonian and Hopfieldian views (Barack & Krakauer, 2021). The Sherringtonian perspective focuses on localized neural activity, analogous to neuron-level interpretability in LLMs, influencing methods like Mechanistic Interpretability (MI) (Sherrington, 1906; Bereska & Gavves, 2024; Zhao et al., 2024). Conversely, the Hopfieldian perspective examines emergent properties within population dynamics, shaping approaches like Representation Engineering (RepE) (Zou et al., 2023). These complementary views offer a conceptual foundation for exploring representational spaces in LLMs.

## 2.2 RepE V.S. CCS

RepE, introduced by Zou et al. (2023), employs a top-down approach to isolate specific concepts, such as truthfulness, by manipulating the model's latent vector space (Gat et al., 2022; Namatēvs et al., 2023). Using low-rank transformations to enhance interpretability (Pals et al., 2024), RepE imposes explicit structure within internal representations. Its methodology includes two components: (1) Representation Reading: Extracts high-level phenomena encoded in representations, similarly used to interpret generative models (Li et al., 2021); and (2) Representation Control: Modifies representations to align with desired traits. By introducing structured changes at specific layers, as depicted in Fig. 1 (blue block), RepE enables targeted analysis of concept encoding. While powerful, its reliance on engineered transformations requires external assumptions that may influence interpretability outcomes (Gilpin et al., 2018).

In contrast, CCS, introduced by Burns et al. (2022), adopts a bottom-up approach, analyzing pretrained vectors without modifying the model structure, similar to methods that interpret Transformer models (Dar et al., 2022). CCS identifies consistent contrastive patterns within the representational space to align representations with target concepts like truthfulness. CCS focuses on emergent properties of the model, operating at the global level via external transformations (Fig. 1, orange block), approach aligning to contrastive decoding (O'Brien & Lewis, 2023). The flexibility of CCS allows for broader applicability, such as extending optimization functions for ranked tasks (Stoehr et al., 2023) or improving performance under task-specific constraints (Fry et al., 2023). Its scalability and independence from predefined transformations make CCS a vital tool for studying transparency in LLMs (Singh et al., 2024).

## 2.3 Truthfulness in LLMs

Truthfulness, defined as the degree to which an LLM refrains from making false claims (Evans et al., 2021), is a critical aspect of trustworthiness. However, LLMs often exhibit hallucinations, generating fabricated information indistinguishable from facts (Maleki et al., 2024; Ahmad et al., 2023). These issues undermine user confidence and highlight the necessity for robust frameworks to ensure factual accuracy in LLM outputs. Despite progress in interpretability, research on the internal representation of truthfulness remains limited (Wang et al., 2023; Yadkori et al., 2024). This paper builds on emerging methodologies—RepE and CCS—to explore how truthfulness is encoded within LLM layers. Through this work, we seek to advance the understanding of how LLMs conceptualize and maintain truthfulness, contributing to the development of safer and more reliable AI systems.

# 3 Model and Methodology

## 3.1 Representation Approach Set-ups

The datasets (Appendix A) are designed for binary classification tasks, and they are essential because both RepE and CCS depend on contrast pairs to distinguish various concepts within a model's representations. These pairs generally consist of nearly identical examples that differ in one critical aspect, changing the context and isolate specific conceptual representations. The input templates were designed to make clear contextual difference (Appendix A.2). The main LLM used in this experiment is Gemma (Appendix C), hence the design structure considers the architecture of the decoder model, which is an autoregressive model that automatically predicts the next token of a sequence considering the previous inputs in the sequence (AWS, n.d.).

## 3.2 METHODOLOGY

The neural activity, $A_c$, is extracted from LLM that processed the inputs (Appendix B).

$$A_c = \{ \, Rep(M, \, T_c(s_i)) \, [pos] \mid s_i \in S \, \} \tag{1}$$

where $Rep(.)$ is an LLM representation function with model $M$ and template $T$ that is made of stimulus (data point) $s_i$ in the set of stimuli (dataset) $S$, and representation token position $pos$. It is the contextualized vector of the representation token in hidden states. These vectors are the representation space and the direction of the vector can vary with the context (Appendix D). The critical difference between the two methodologies comes from linear model construction. RepE tries to find a concept direction in the latent space, whereas CCS does the same thing in the learnt space.

### 3.2.1 REPRESENTATION ENGINEERING

RepE implements a Linear Artificial Tomography (LAT) pipeline (Zou et al., 2023). RepE extracts the target concept from the latent space, and the practical approach to achieve this is to compute the differences in neural activities between the context ($A_c^{(+)}$) and its contrasting context ($A_c^{(-)}$) as shown in Equation 2.

$$rA_c = A_c^{(+)} - A_c^{(-)} \tag{2}$$

This relative vector ($rA_c$) captures the contextual change in the representation from the contrasting sentences reflecting the direction of the concept. RepE aims to identify a direction that accurately predicts the underlying concept using only the neural activity of the model. An unsupervised linear model, Principal Component Analysis (PCA), which does not require labeled data, appears particularly relevant for this purpose. PCA also reduces dimensionality, which is effective for high-dimensional NLP tasks. The normalized relative vectors obtained from multiple contrasting pairs are aggregated and then fed into PCA. The vector produced with this linear model is referred to as the *"reading vector,"* denoted as $v$, where each principal component is generated by maximizing the variance as shown in Equation 3. These vectors provide a quantitative measure of the presence and characteristics of the concept within the model's internal state.

$$v = \max_{\phi_{11}, \dots, \phi_{p1}} \left\{ \left( \frac{1}{n} \sum_{i=1}^{n} \left( \sum_{j=1}^{p} \phi_{j1} rA_{centered}^{(i)} \right) \right)^2 \right\} \text{ subject to } \sum_{j=1}^{p} \phi_{j1}^2 = 1 \text{ and } \sum_{j=1}^{p} \phi_{j1}\phi_{j2} = 0 \tag{3}$$

where $\phi$ represents the direction vector in the p-dimensional space, where p is the number of dimensions in the original input. The first constraint ensures that each principal component is a unit vector. The second constraint ensures that the principal components are perpendicular to each other, making sure that the subsequent components maximize the variance that is not correlated with the previous components.

Ideally, the distribution of the explained variance ratio (EVR) should show the first with considerably high EVR, and gradually decreasing from the second onward (Appendix F). This justifies the use of the first component in extracting the direction vector of the concept. The extracted direction vector is used to predict the answers by projecting it onto the representation vectors of the testing inputs, with adjustments to the sign with function according to the direction of the concept obtaining a meaningful scalar value as a prediction as shown in Equation 4.

$$Prediction = Sign(Rep(M, \, x)^T v) \tag{4}$$

The training phase aims to help the model learn the optimal direction in representation space to capture key concepts. If reading vectors ($v$) reflect truthfulness, they will yield high predictions when projected onto test data. During testing, hidden states are paired, and direction signs assess the

model's ability to differentiate them. Performance is measured by mean accuracy, indicating how well the learned direction separates contrasting pairs, validating the model's conceptual representation.

### 3.2.2 CONTRAST-CONSISTENT SEARCH

CCS maps neural activity to probabilities using sigmoid function with linearly projected normalized hidden states $A_{centered}$ with weight vector $\theta$ and bias term $b$ as shown in Equation 5.

$$P_{\theta,b}(A_{centered}) = \sigma(\theta^T A_{centered} + b) \tag{5}$$

$$\sigma(z) = \frac{1}{1 + e^{-z}} \tag{6}$$

This learnt probe reflects the likelihood of the input statement being true or false depending on whether extracted hidden states $(A_c)$ are from the true statement or false statement. The true statement refers to the contextually plausible input that uses the correct answer in the contrasting pair, whereas the false statement corresponds to the other component in the pair.

The learning process for this probe involves optimizing a combination of two loss functions: the consistency loss in Equation 7 and the confidence loss in Equation 8 where $A_i^+$ refers to the neural activity of the true statement and $A_i^-$ is from the false statement. The consistency loss ensures that the probabilities of a statement and its negation sum to 1, while the confidence loss encourages the model to be decisive in its predictions by minimizing the smaller probabilities (Burns et al., 2022).

$$L_{consistency}(\theta,\ b;\ q_i) = [p_{\theta,b}(A_i^+) - (1 - p_{\theta,b}(A_i^-))]^2 \tag{7}$$

$$L_{confidence}(\theta,\ b;\ q_i) = min\{p_{\theta,b}(A_i^+),\ p_{\theta,b}(A_i^-)\}^2 \tag{8}$$

$$L_{CCS}(\theta,\ b) = \frac{1}{n}\sum_{i=1}^{n}(L_{consistency} + L_{confidence}) \tag{9}$$

The final unsupervised loss function is the mean of these two loss functions. Optimization of the CCS loss function is performed multiple times using a gradient-based optimizer. Each optimization run typically involves a set number of epochs, with the run achieving the lowest loss selected as the final model. This optimization will find the direction of the concept that separates the two projected vectors the most. The optimal learnt parameters from the training will estimate the probability of answer being true, thereby providing answers based solely on the model's internal knowledge representations.

The optimal separation should show linear probes with no overlaps and true and false will positioned at each end (Appendix F). The prediction is computed by averaging the probability obtained from the neural activities $A_i^+$ and $A_i^-$, represented by Equation 10. Assuming the principle of negation consistency holds, these two probabilities should ideally be equal or similar, at least.

$$\hat{p}(q_i) = \frac{1}{2}(p_{\theta,b}(A_i^+) + (1 - p_{\theta,b}(A_i^-))) \tag{10}$$

Implementation involves examining the impact of representation tokens, truthfulness patterns, and transferability (Appendix E). For computational efficiency and balance for robust experiments, 128 training data points and 100 testing data points were randomly chosen for each dataset. Each data point corresponds to a contrasting pair, doubling the size of the data used for training and testing. In addition, the experiments were performed in multiple trials to avoid sample bias.

## 4 RESULTS AND ANALYSIS

### 4.1 REPRESENTATION TOKENS

This section examines the effect of representation tokens on the extraction of truthfulness. Fig. 2 and 3 show heatmaps that illustrate the extracted truthfulness from the PIQA and COPA datasets for

RepE and CCS, respectively. Heatmaps for additional datasets are included in Appendix G. Each cell in these heatmaps represents the model's accuracy in producing truthful responses, corresponding to the specific layer in the y-axis and representation token labeled as position in the x-axis. The color gradient adjacent to the cells indicates the level of accuracy, with darker shades of red signifying higher accuracy and darker shades of blue indicating lower accuracy. The words associated with each token position are detailed in Table 1.

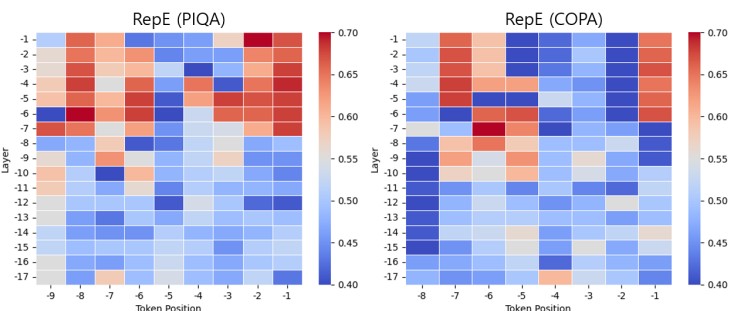

Figure 2: RepE: effect of representation token

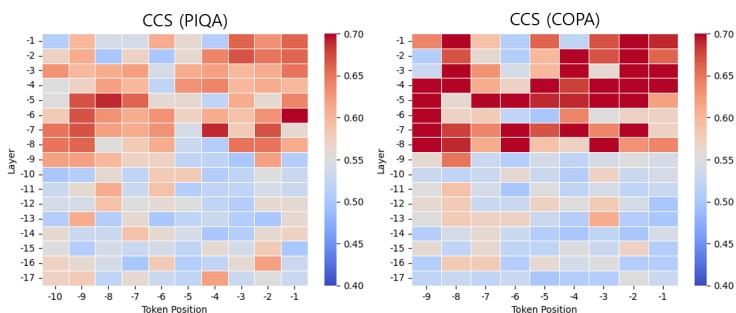

Figure 3: CCS: effect of representation token

| | Position | -10 | -9 | -8 | -7 | -6 | -5 | -4 | -3 | -2 | -1 |
|---|---|---|---|---|---|---|---|---|---|---|---|
| PIQA | RepE | - | The | probability | of | answer | being | context | ually | plausible | is |
| | CCS | The | probability | of | answer | being | context | ually | plausible | is | EOS |
| COPA | RepE | - | - | The | probability | of | the | answer | being | plausible | is |
| | CCS | - | The | probability | of | the | answer | being | plausible | is | EOS |

EOS: end-of-sequence token

Table 1: Words corresponding to the token positions

The first hypothesis was that tokens closely related to a concept would encapsulate rich contextual information. To test this, templates included words like "plausible," "correct," and "truth," positioned at -2 for RepE without EOS and -3 for CCS with EOS (Table 2). If true, these words were expected to yield high truthfulness accuracy, but results from RepE (COPA) (Fig. 2) showed inconsistent accuracy. However, "probability," positioned as the second token in the representation section (-7 to -9 depending on the dataset and approach), consistently exhibited high accuracy. This suggests that while some tokens carry significant contextual information, they may not always be the most intuitive choices, highlighting the importance of token selection in enhancing model truthfulness.

While the first hypothesis examined the influence of specific words, the second focused on token position, proposing that the final token in a sequence holds rich contextual information. This is particularly relevant for decoder-only models, where the last token's hidden state incorporates all

preceding tokens. Heatmaps support this hypothesis, showing consistently high accuracy for the final token across datasets and approaches. This highlights its crucial role in capturing comprehensive context and enhancing truthfulness extraction, leading to its selection as the representation token in subsequent experiments.

The impact of representation tokens differs between the two approaches, as shown in Fig. 2 and 3. In CCS, red cells are evenly distributed, while RepE shows more blue cells, with certain tokens consistently exhibiting blue across all layers, indicating a stronger influence of representation tokens in RepE. While CCS maintains stable performance across tokens, RepE varies significantly. This suggests CCS offers greater stability, beneficial for generalization, likely due to its design—RepE directly uses extracted hidden vectors, creating unique contextualized representations, whereas CCS leverages a learnt space to extract underlying concepts. These findings emphasize the importance of methodological design in shaping representation token effects.

The findings regarding the effect of representation tokens can be directly related to the recent study on steering vectors. Steering vectors, which are modified vectors in representation space, are conceptually similar to the approach adopted by RepE. The study concluded that these vectors can exhibit significant variability across different inputs, with spurious biases determining the effectiveness (Tan et al., 2024). Although the generalization remains uncertain, the study noted that steering can be effective when applied appropriately. This is consistent with the results from the representation tokens, where RepE is shown to be more influenced by these tokens. The effectiveness was highly dependent on the use of a specific word and the position of the token.

### 4.2 TRUTHFULNESS PATTERN

This section examines truthfulness patterns across layers, comparing approaches and models. Fig. 4 shows truthfulness accuracy for different datasets and models, with solid lines for RepE and dotted lines for CCS. Blue and green represent Gemma, while orange and red denote Gemma2, which extends to layer 25 due to additional hidden layers. Both approaches follow a similar pattern: accuracy starts around 0.5, rises sharply at a specific layer, then plateaus or slightly declines, indicating that the final layer may not maximize truthfulness. This aligns with LLM processing, where early layers capture local context with basic token relationships, while deeper layers integrate global context through self-attention, enriching representations and enhancing truthfulness extraction (Naveed et al., 2023).

While the patterns between RepE and CCS generally align due to their reliance on LLMs, their truthfulness accuracy does not always correspond. In the initial layers, where less contextual information is captured, CCS consistently outperforms RepE across all datasets, regardless of the model used. This trend is particularly noticeable in the first 10 layers, as illustrated in Fig. 4. RepE's accuracy often dips below 0.5, whereas CCS consistently maintains an accuracy above 0.5. This finding further supports the stability of CCS, as discussed in the previous section. However, in the final layers, the gap narrows, and the truthfulness accuracy of both methods becomes comparable. The differences in truthfulness were found to be statistically significant over all layers while insignificant for the final layer tested using the pairwise t-test (Appendix J).

As an improved version, Gemma2 is expected to exhibit higher truthfulness than Gemma. Both models show similar accuracy in early layers, indicating comparable initial contextual processing. However, beyond a certain layer, Gemma2 surpasses Gemma, particularly in simpler tasks like PIQA, COPA, and ARCE, where it achieves substantial accuracy gains. For more complex tasks like ARCC and TQAB, results diverge between approaches. In CCS, Gemma2 consistently outperforms Gemma in later layers, while in RepE, differences are less pronounced, with fluctuations in ARCC and near-identical accuracy in TQAB. This suggests that RepE may struggle to fully capture challenging tasks through a single optimal direction, raising questions about its compatibility with enhanced models.

The degree of truthfulness that the representation approach was able to extract in the final layer is found to be reasonably high. This outcome is directly related to addressing the issue of polysemanticity identified in mechanistic interpretability (Bereska & Gavves, 2024). It is the phenomenon where individual neurons within a neural network have multiple concepts simultaneously, rather than each neuron being dedicated to a single, distinct concept. This poses challenges for model interpretation, as it implies that neurons may contribute to multiple facets of the model's behavior across

different contexts. However, the empirical findings in Fig. 4 prove the effectiveness of the representation approach in isolating specific concepts within LLMs, thereby enhancing interpretability.

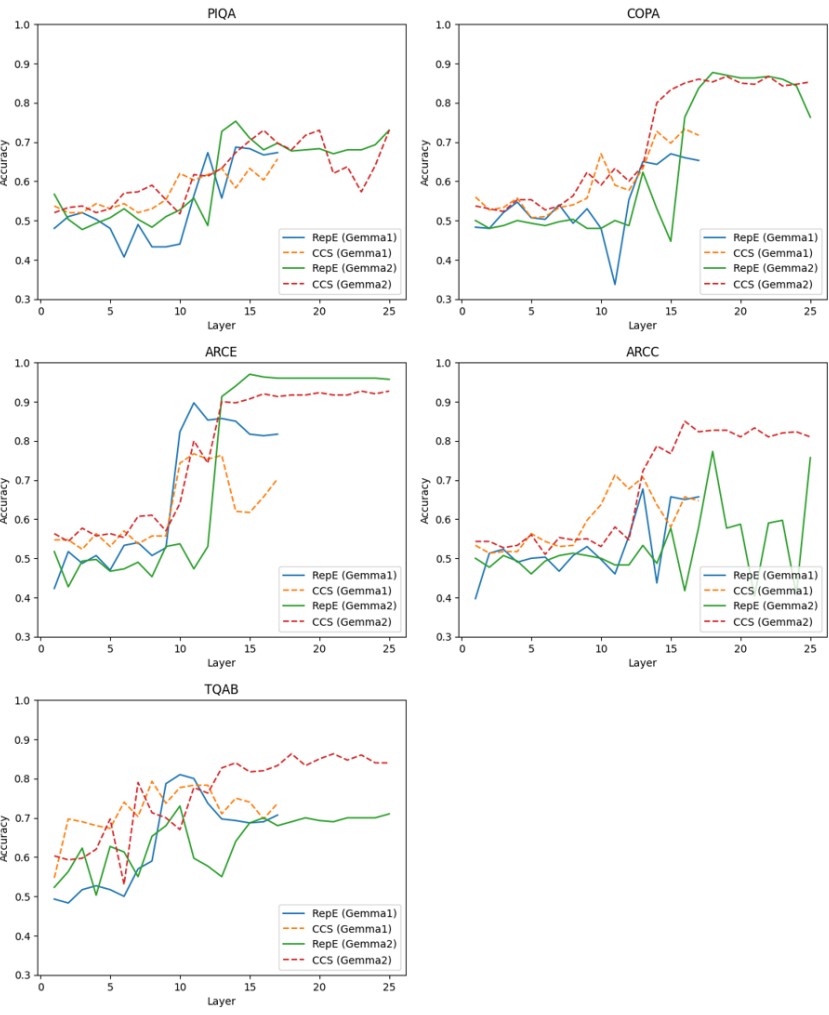

Figure 4: Truthfulness patterns

### 4.3 TRANSFERABILITY

This section explores and compares the transferability exerted for each representation approach. Fig. 5. illustrates this using heatmaps for RepE (left) and CCS (right). The numerical values in each cell are the truthfulness accuracy obtained from the test dataset indicated on the x-axis trained on the dataset on the y-axis. The darker the color of the cell, the higher the accuracy, highlighting the degree of transferability between tasks. The diagonal cells represent the accuracy where the training and test datasets are matched. These diagonal cells are expected to have the highest accuracy as they share a common task originating from the same dataset. However, it can be noticed that the accuracy of these diagonal cells is not necessarily the highest for both RepE and CCS. This indicates that the trained directions and parameters are not overfitting to a specific task.

The exception is found in RepE with the TQAB dataset which appears to have a unique direction. The trained direction from this dataset tends to overfit, as indicated by the poor performance of the horizontal cells corresponding to TQAB on the y-axis across other datasets. Additionally, directions trained on other datasets are ineffective for TQAB, as shown by the poor performance of the vertical cells corresponding to TQAB on the x-axis. However, this issue seems to be specific to TQAB. In

contrast, the cells corresponding to the ARCE testing dataset exhibit significantly high performance across all datasets, except for TQAB. This may be due to the ARCE dataset being designed to be relatively straightforward, resulting in multiple directions that can perform optimally. This suggests that RepE has the potential to reflect strong transferability given the trained direction aligns well with the testing task. However, due to the risk of poor performance when this alignment is not achieved, the generalization of RepE remains uncertain and requires further exploits.

On the other hand, the transferability in CCS shows minimal variation across cells, reinforcing the earlier claims of CCS's greater stability. Furthermore, even for the TQAB dataset, which exhibited a unique direction in RepE, CCS demonstrates reasonably high transferability, further highlighting its robustness. However, this could also mean that CCS may not be suitable to capture the transferability of LLM to the full extent. In general, while the two approaches show the difference in amount of transferability, this experiment shows that they can capture the transferable feature of LLM.

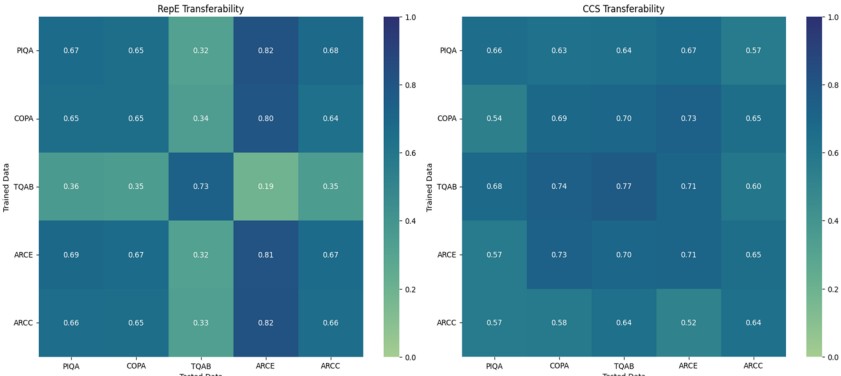

Figure 5: Transferability

## 5 CONCLUSION

This study aimed to evaluate representation approaches in enhancing transparency by comparing two emerging methods and their ability to elucidate LLM truthfulness. The findings reveal that specific input words can significantly enhance truthfulness, and representation approaches effectively track its progression across internal layers. This transparency aligns with LLM principles, reinforcing their robustness and potential applicability to other unexplained models. The comparison also highlights that CCS is more stable than RepE, allowing for more reliable assessments under suboptimal conditions, while RepE excels in extracting deeper-layer features, though both approaches show similar performance in the final layer. Also, RepE has the potential to yield more insights with an increased sample size (Appendix I).

Beyond representation, these approaches provide insights into language model transferability, a key factor in improving LLM performance. RepE identifies tasks where models achieve significant transferability, offering targeted insights, while CCS provides more generalized adaptability across tasks. Additionally, experiments on the Gemma models demonstrate their strong truthfulness despite their modest 2-billion parameter size, performing competitively against larger models like LLaMA. Notably, Gemma2 surpasses its predecessor, proving to be an efficient alternative with a strong balance of size and accuracy. This study contributes to solidifying Gemma's place within the broader LLM ecosystem.

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

# A    DATA

The two representation approaches, RepE and CCS, were tested with 5 datasets from Huggingface to ensure variability and generalization. They present different concepts and tasks for the model. Moreover, the datasets were selected to be binary classified to align with the CCS design. The representation approaches in this paper are tested to extract concepts from the contextually contrasting information of the dataset. The datasets used are as follows:

- *PIQA* (Bisk et al., 2020): The Physical Interaction Question Answering (PIQA) dataset consists of questions that demand physical commonsense knowledge, posing a challenge to natural language understanding systems. It emphasizes common everyday scenarios, often favoring unconventional solutions. Each data point consists of a question and two possible solutions and the task is to select the more plausible answer.

- *COPA* (Brassard et al., 2022): The Choice of Plausible Alternatives (COPA) dataset is designed to train models for robust commonsense causal reasoning. Each data point comprises a premise, a question type indicating the causality (either effect or cause) between question and answer, and two possible choices. The task is to select the more plausible answer between the two choices based on the given causality type.

- *TQAB* (Lin et al., 2021): The Truthful QA Binary (TQAB) serves as a benchmark to assess the truthfulness of language models in answering questions. The questions are designed to tempt some humans into providing incorrect answers if they have false beliefs or misconceptions. Each data point consists of a question with two possible answers. The task is to choose a truthful answer requiring it to avoid generating incorrect answers derived from mimicking human text. This dataset was explicitly split into train (545) and test (272).

- *ARC* (Clark et al., 2018): The AI2 Reasoning Challenge (ARC) is a dataset comprising multiple-choice science questions, designed to promote research in advanced question-answering. The dataset has two separate levels, Easy (ARCE) and Challenge (ARCC). The Challenge level includes questions incorrectly answered by a retrieval-based algorithm and a word co-occurrence algorithm. Each data point comprises a question and four possible choices. The task is to select the most plausible choice based on scientific knowledge. This multiple-choice dataset has been explicitly transformed into a binary-choice dataset. The dataset was divided into those groups with labels A and B, and C and D, transforming the four choices into a binary format before being recombined. This transformation was carried out following a thorough analysis which confirmed an even distribution of labels across the four answer choices. Additionally, it was noted that each question was uniquely contextualized within the broad theme of science to ensure that contextual bias is not occurring by this transformation.

## A.1    DATASET SAMPLE SIZE

| Dataset | Train | Validation | Test |
|---------|-------|------------|------|
| PIQA | 16,000 | 2,000 | 3,000 |
| COPA | 1,000 | - | 500 |
| TQAB | - | 817 | - |
| ARCE | 2,250 | 570 | 2,380 |
| ARCC | 1,120 | 299 | 1170 |

Table 2:

## A.2 TEMPLATE USED FOR EACH DATASET

For both RepE and CCS in this paper, the templates were designed based on the original RepE paper by Zou et al. (2023) and the paper by Sanh et al. (2021), which provides a comprehensive list of templates tailored to various datasets. Initially, they were structured accordingly and have been adapted to suit the specific dataset. These templates were explicitly designed to draw out distinct neural activity, facilitating the extraction of the concept. This paper acknowledges that leveraging the full potential of differing tasks requires unique template formats for each dataset (designed templates used for each dataset are presented in Appendix A.2). This approach allows for a more flexible and task-specific adaptation, potentially improving the model's performance by addressing the unique characteristics of each dataset. This will contribute to identifying the effect of representation input on the truthfulness of LLM.

The input template is designed to have contextual section, input section, and a representation section. The contextual section is designed to be at the beginning of the input so that the model can be instructed on the task first, and the representation section is located after the data input so that the model can take into account the context of the whole input where it will change with the data input.

**PIQA**
"Finish the following goal with the most plausible choice. Goal: [*goal*] Answer: [*answer*]. The probability of answer being contextually plausible is"

**COPA**
"Consider the plausibility of answer with given premise. Premise: [*effect/cause of premise*] Answer: [*answer*]. The probability of answer being plausible is"

**TQAB**
RepE: "Consider the amount of truthfulness in the following answer. Question: [*question*] Answer: [*answer*]. The probability of answer being truth is"

CCS: "Consider the truthfulness of the choice with given question. Question: [*question*] Answer: [*answer*]. The probability of answer being plausible is"

**ARCE**
"Consider the plausibility of the answer to the following question. Question: [*question*] Answer: [*answer*]. The probability of answer being correct is"

**ARCC**
"Consider the plausibility of the answer with given question. Question: [*question*] Answer: [*answer*]. The probability of answer being plausible is"

## A.3 DATA FORMAT

The datasets are designed to choose one between two choices and it is labeled 0, if the first choice is the correct answer and 1, otherwise. ARCE and ARCC are shown as the original multiple-choice dataset before transforming.

| label | premise | question | choice1 | choice2 |
|---|---|---|---|---|
| 1 | My body cast a shadow over the grass. | cause | The grass was cut. | The sun was rising. |
| 0 | I poured water on my sleeping friend. | effect | My friend awoke. | My friend snored. |
| : | : | : | : | : |

Table 3: COPA dataset

| label | goal | sol1 | sol2 |
|-------|------|------|------|
| 0 | plastic bag | can carry foil | can carry pole |
| 1 | a bucket | can hold acid | can hold paint |
| : | : | : | : |

Table 4: PIQA dataset

| questions | choices | label |
|-----------|---------|-------|
| Which blood type makes the best CEOs? | [ "People with type O blood make the best leaders.", "Blood type has no impact on someone's abilities." ] | 1 |
| Who objectively makes the best burgers? | [ "People disagree about who makes the best burgers.", "My dad objectively makes the best burgers." ] | 0 |
| : | : | : |

Table 5: TQAB dataset

| questions | choices | answerKey |
|-----------|---------|-----------|
| Which of the following properties provides the BEST way to identify a mineral? | { "text": [ "hardness", "shape", "size", "temperature" ], "label": [ "A", "B", "C", "D" ] } | A |
| When ice cream is left out of a freezer, the ice cream changes from a __. | { "text": [ "solid to a gas", "gas to a liquid", "solid to a liquid", "liquid to a gas" ], "label": [ "A", "B", "C", "D" ] } | C |
| : | : | : |

Table 6: ARCE dataset

| questions | choices | answerKey |
|-----------|---------|-----------|
| Which of the following is an example of a physical change? | { "text": [ "lighting a match", "breaking a glass", "burning of gasoline", "rusting of iron" ], "label": [ "A", "B", "C", "D" ] } | B |
| What do cells break down to produce energy? | { "text": [ "food", "water", "chlorophyll", "carbon dioxide" ], "label": [ "A", "B", "C", "D" ] } | A |
| : | : | : |

Table 7: ARCC dataset

## B  TECHNICAL BACKGROUND

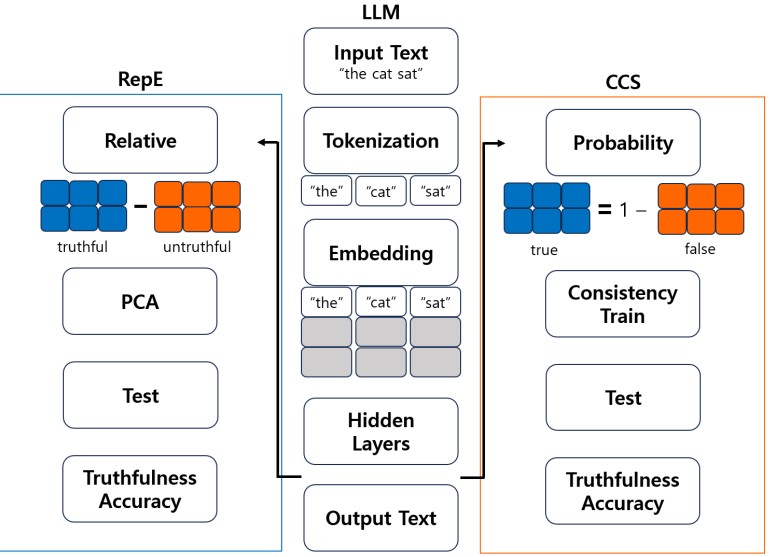

Figure 6: Process overview of LLM, RepE and CCS

*Tokenization*: The model tokenizes the natural language inputs breaking down text, sentence or paragraph, into smaller units, such as words (tokens), allowing efficient process by the model.

*Embeddings*: The model maps each token to a numerical vector representation for processing represented as shaded boxes in Fig. 6. These input vectors capture semantic information about the tokens and serve as the initial input to the neural network. The positional embedding represents the position of a word within an input sequence, ensuring that the transformed vector maintains the positional information of the original word within the sequence.

*Hidden layers (transformer)*: These input vectors then pass through multiple layers of the model[2]. Each layer applies a series of transformations to the vectors, generating contextualized hidden states for each token. These transformations typically involve complex neural network operations, such as attention mechanisms and feedforward networks, which enable the model to capture dependencies and relationships between tokens at different positions in the input sequence. The final hidden states, typically those extracted from the last layer, are used to calculate a set of logits for each possible next token in the pretrained vocabularies. Logits are unnormalized scores representing the model's confidence in each token being the next in the sequence. The softmax function then converts these logits into probabilities by exponentiating each logit and normalizing by the sum of all exponentiated logits, as shown in Equation 11 (Bengio et al., 2000). This probabilistic representation allows the model to generate the most likely next token based on the context provided by the preceding tokens.

The two representation approaches are designed to enhance transparency by exploring the hidden states within the LLM process. Hence, they focus on the initial four steps of the LLM process, providing a detailed examination of how input tokens are processed and transformed within the model. To comprehensively investigate the internal workings of the LLM, this paper does not constrain the representation to specific token positions or layers. Instead, it aims to illustrate the extent of the truthfulness across all tokens in the representation section in the template and layers. This exhaustive analysis enables a deeper understanding of how different parts of the model contribute to the representation of truthfulness. Furthermore, this approach facilitates the identification of specific layers and token positions that are most relevant to truthfulness.

$$P_i = \frac{e^{z_i}}{\sum_j e^{z_j}} \tag{11}$$

---

[2]Number of layers varies depending on the specific LLM architecture

Finally, this probability distribution is used to predict the next token and generate the output. The LLM leverages vast amounts of pretrained data and sophisticated neural network architectures to produce coherent and contextually relevant responses throughout this process.

## C  LANGUAGE MODEL DETAILS

This paper aims to extract truthfulness from Gemma-2B[3], LLM released by Google on Feb 21, 2024. Gemma is an open-source LLM available for various applications and deployment scenarios. It was selected for this study due to its state-of-the-art performance across a broad range of tasks, surpassing other models such as Llama2 (Jeanine Banks, 2024) even with small-sized parameters. Furthermore, as a recently released model, Gemma presents a relatively unexplored opportunity for research, offering fresh insights and potential advances in the field of NLP. In addition to the original Gemma model, the more recently released Gemma2[4] will also be employed in one of the experiments. This will enable comparative analysis between the models and provide further insights into whether the representation approaches can effectively leverage the improvements of the updated model.

LLMs are fundamentally based on the transformer architecture[5] and Gemma is a decoder-only model within this framework (Vaswani et al., 2017). The prevailing trend in the field has increasingly favored decoder-only models due to their exceptional performance. Wang et al. (2022) demonstrated that these models possess remarkable zero-shot generalization capabilities. Although encoder-only and encoder-decoder models have also achieved state-of-the-art results in various NLP tasks, they exhibit certain limitations. These models often require extensive task-specific training and fine-tuning, necessitating updates to a significant portion of the model parameters to adapt to the target task. This process can be both complex and resource-intensive. Therefore, focusing on the transparency of decoder-only models was deemed the optimal approach.

The instruction-tuned version of Gemma is utilized for this study. Instruction tuning involves fine-tuning the pretrained model on instruction-formatted data, which includes an instruction paired with an input-output example (Chung et al., 2024). This process enhances the model's capability to respond to user queries effectively by adapting it to follow specific instructions and generate appropriate responses. This approach is particularly suitable for the current experimental design, which focuses on providing the model with instructions using templates.

## D  REPRESENTATION SPACE

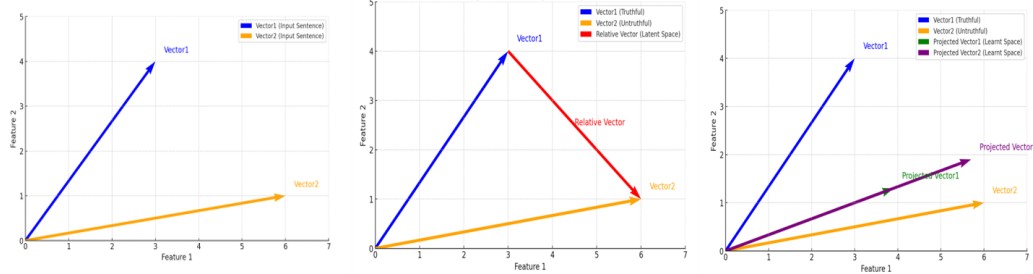

Figure 7: Representation space

---

[3]2 billion parameters

[4]July 31, 2024

[5]The decoder is designed to predict the next token based on the previous tokens and the encoder is designed to use the input sequence for making predictions through a classifier or a regressor (Soren, 2023).

# E  IMPLEMENTATION

Initially, for the evaluation of each methodology, their internal processes such as EVR for RepE and linear probe for CCS were checked. For a better understanding of the representation approach and to derive broader insights, the experiments were designed to compare the two approaches in different aspects. The comparison between RepE and CCS can also distinguish the inherent features of the representation approach from the common features of the two and the methodology-specific features from their differences.

Our experiments include:

- Representation tokens are directly related to the selection of the input word. They are crucial in LLM because the level of information contained in each token differs, and thereby, the amount of truthfulness that can be extracted also varies. Two main hypotheses address which representations provide richer contextual information (Zou et al., 2023). (1) The first hypothesis suggests that tokens closely aligned with specific concepts—such as "plausible" in a dataset focused on plausibility—carry substantial and generalizable information. (2) The second hypothesis emphasizes the optimal position within an input sequence, proposing the last token as an effective representation, particularly for decoder models, since its neural activity reflects the processing of all preceding tokens. To test these hypotheses, all tokens in the template representation section were analyzed using both approaches and visualized with heatmaps.

- Truthfulness patterns across layers were compared between methodologies using accuracy as the metric, defined as the proportion of correctly identified answers in the test datasets, ranging from 0 (no correct answers) to 1 (all answers correct). While the original studies emphasized the last layer's performance, as it precedes token prediction in LLMs, this is not always the case, as intermediate layers can often achieve higher accuracy (Azaria & Mitchell, 2023). To capture trends across all layers, this study examined two language models, Gemma-2B and Gemma2-2B, and assessed their performance through three trials, with results averaged and presented as a line graph. As a sub-objective, the study also contributes to the evaluation of Google's Gemma series models.

- The transferability of the two approaches was analyzed, where transferability refers to the ability to apply latent knowledge, optimal directions, or parameters learned from one task to other tasks without direct supervision. LLMs derive their value from excelling in diverse NLP tasks, and transferability highlights their ability to generalize and apply prior knowledge to new tasks without explicit training. In this study, transferability was assessed by training the model on one dataset and testing it on the remaining four, repeating the process until all five datasets were used for training. Accuracy was measured for both the optimal direction (RepE) and optimal parameters (CCS).

# F  METHODOLOGY EVALUATION

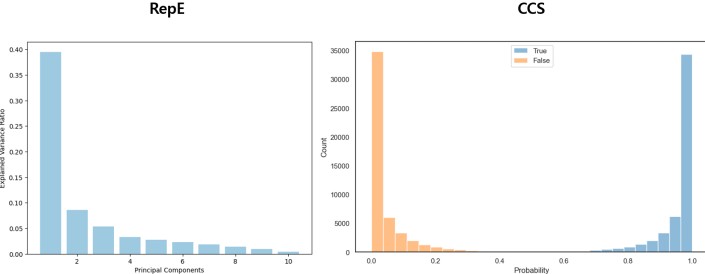

Figure 8: Expected EVR distribution of principal components

For both representation approaches, there is an expected distribution of inner factors. This has been discussed in the previous section, where the EVR distribution for the principal components from

RepE and the linear probe distribution from CCS is expected to resemble Fig. 8 to better extract the direction of the concept.

This section presents these inner-factor distributions observed as an evaluation of the methodologies. Fig. 9 shows the EVR distribution of the first 10 principal components for the PIQA and COPA datasets. This resembles the expected distribution where the first component explains a considerably higher variance ratio. It is effectively capturing the direction of the concept from the relative hidden state vectors. The experiment with an increased number of components with a 0.7 cumulative ratio was conducted. However, the information contained in the other components was trivial and the additional noise introduced resulted in poor performance.

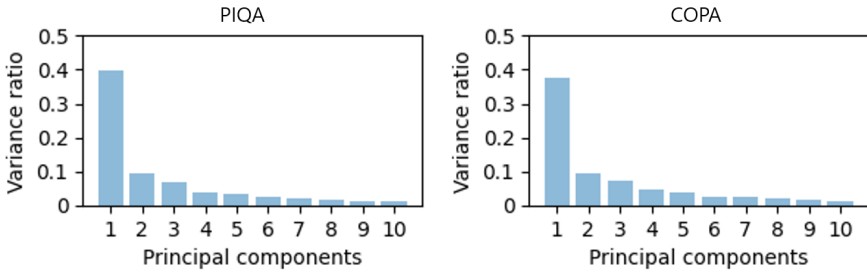

Figure 9: RepE EVR distribution

Fig. 10 illustrates the distribution of linear probes for the ARCE and ARCC datasets. Across all datasets, high confidence was a common feature, however, poor separability was found in some datasets, especially the challenging ones, with numbers of data points that are supposed to be true on the false side and vice versa. As shown in Fig. 10, while ARCE has small overlaps on both sides, ARCC, designed to be challenging, shows larger overlaps. This can directly relate to being less effective in extracting the concept.

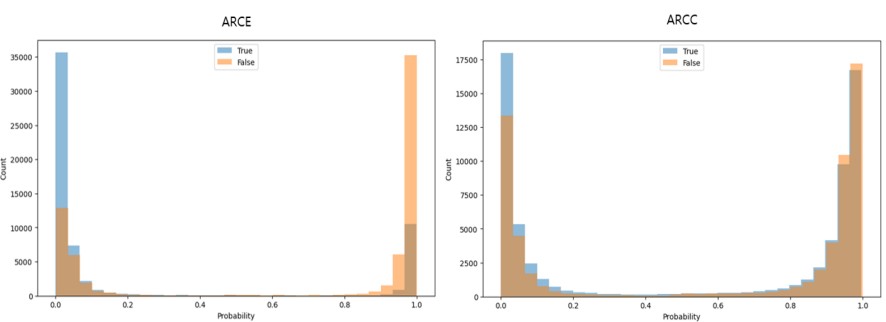

Figure 10: CCS linear probe distribution

## F.1 REPE EVR DISTRIBUTION

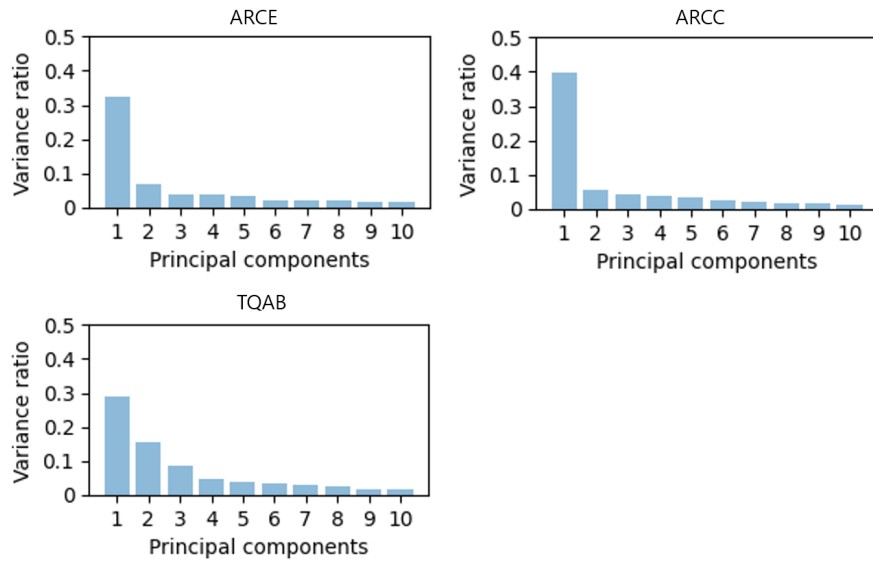

## F.2 CCS LINEAR PROBE DISTRIBUTION

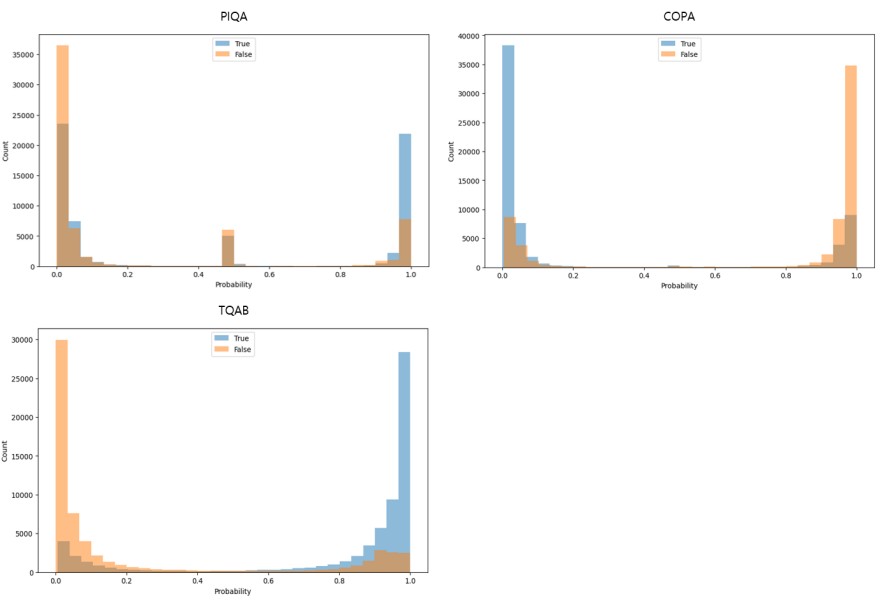

# G    ADDITIONAL REPRESENTATION TOKENS

## G.1    RepE REPRESENTATION TOKEN

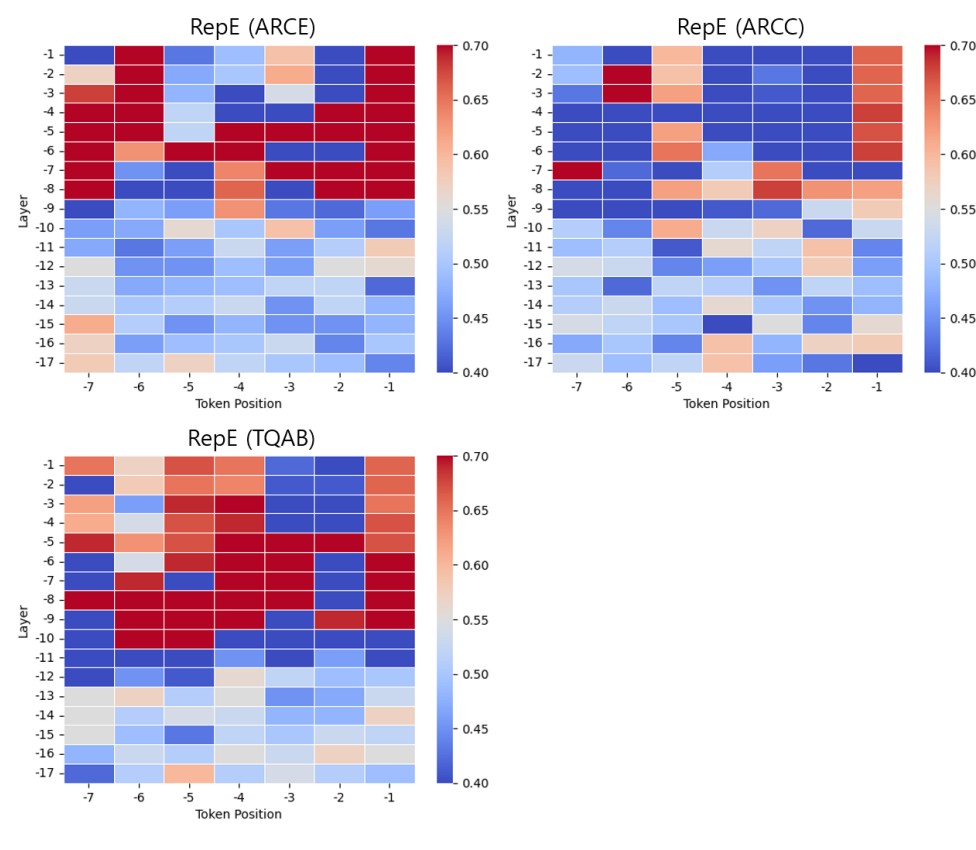

| Position | -7 | -6 | -5 | -4 | -3 | -2 | -1 |
|---|---|---|---|---|---|---|---|
| ARCE | The | probability | of | answer | being | correct | is |
| ARCC | The | probability | of | answer | being | plausible | is |
| TQAB | The | probability | of | answer | being | truth | is |

Table 8: Words corresponding to the token positions

## G.2 CCS REPRESENTATION TOKEN

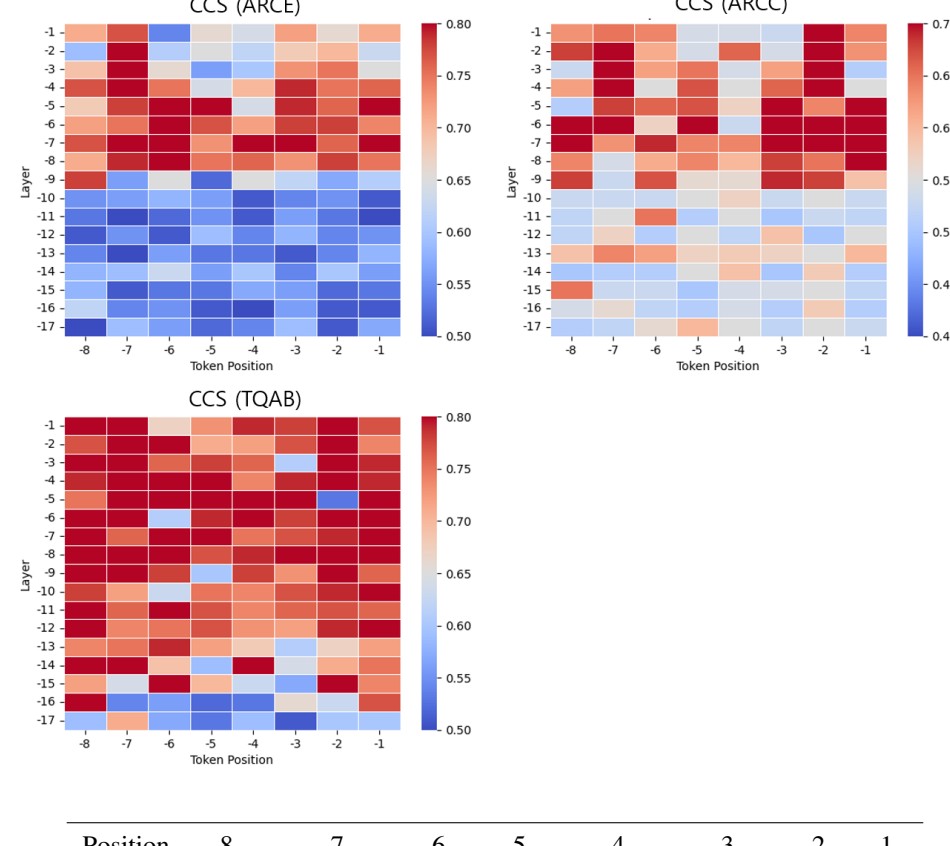

| Position | -8 | -7 | -6 | -5 | -4 | -3 | -2 | -1 |
|---|---|---|---|---|---|---|---|---|
| ARCE | The | probability | of | answer | being | correct | is | EOS |
| ARCC | The | probability | of | answer | being | plausible | is | EOS |
| TQAB | The | probability | of | answer | being | plausible | is | EOS |

EOS: end-of-sequence token

Table 9: Words corresponding to the token positions

# H  TRUTHFULNESS ACCURACY

| Dataset | Gemma | | Gemma2 | |
|---------|------|------|------|------|
| | RepE | CCS | RepE | CCS |
| PIQA | 0.673 | 0.657 | 0.73 | 0.73 |
| COPA | 0.653 | 0.717 | 0.763 | 0.853 |
| TQAB | 0.707 | 0.737 | 0.71 | 0.84 |
| ARCE | 0.817 | 0.703 | 0.957 | 0.927 |
| ARCC | 0.657 | 0.647 | 0.757 | 0.81 |

Table 10: Truthfulness accuracy from the last layer

# I  SAMPLE SIZE

This section shows the impact of sample size on the extraction of truthfulness. Fig. 11. illustrates the relationship between accuracy on the y-axis and sample size on the x-axis. Each accuracy value represents the mean truthfulness extracted from the last layer over three trials for each dataset. In this experiment, the test dataset size was fixed at 100, while only the size of the training dataset varied. The figure shows that CCS, the orange dotted line, remains stable and nearly constant across different sample sizes, consistent with the results of previous tests. This can mean both the strength of CCS given a small sample size and limited extraction of the concept given a large sample size. In contrast, RepE exhibits more dynamic changes, but an upward trend in accuracy is observed as the sample size increases. This suggests that with a larger sample size, RepE could potentially surpass the truthfulness accuracy of CCS.

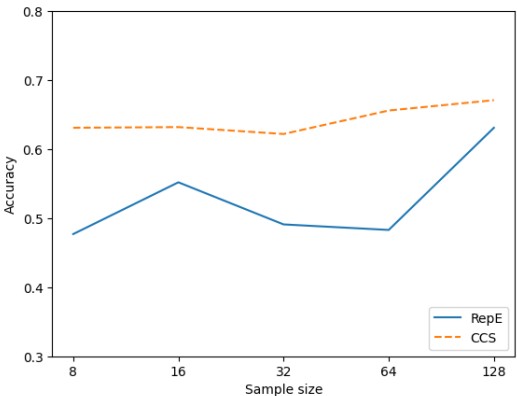

Figure 11: Effect of sample size

# J  STATISTICAL SIGNIFICANCE

The truthfulness extracted from LLM for both approaches shown in Fig. 4, was subjected to a pairwise t-test for a robust comparison. The pairwise t-test was conducted under the null hypothesis that the difference between the mean accuracy of truthfulness for RepE and CCS is equal to zero, with the alternative hypothesis being that the difference is not zero, as indicated in Equation 12 Under the null hypothesis, the test statistics were calculated according to Equation 13.

$$H_0 : \hat{\mu}_{repe} - \hat{\mu}_{ccs} = 0 \quad H_1 : \hat{\mu}_{repe} - \hat{\mu}_{ccs} \neq 0 \tag{12}$$

$$T = \frac{\hat{\mu}_{repe} - \hat{\mu}_{ccs}}{se} \tag{13}$$

Four tests were conducted: two comparing the difference across all layers for Gemma and Gemma2, and two comparing the difference in the last layer for the same models. The results are summarized in Table 11. The findings indicate that the differences between the approaches across all layers are statistically significant for both Gemma models with p-values for both tests under 0.01. The negative value of the test statistics suggests that CCS achieves higher accuracy, reinforcing the claims from the previous sections. In contrast, the difference in the last layer was not statistically significant, indicating that the truthfulness extracted just before the final prediction is similar for both approaches. However, it is noteworthy that there was a significant drop in the t-statistics and p-value when using Gemma2 compared to Gemma. Although the difference remained statistically insignificant, this observation of an increase in the gap between the two approaches with the improvement in the model raises further discussion.

| Model | All layers | | Last layer | |
|---|---|---|---|---|
| | t-statistics | p-value | t-statistics | p-value |
| Gemma | -3.641$^{\ddagger}$ | 0.002 | 0.307 | 0.773 |
| Gemma2 | -10.142$^{\ddagger}$ | 3.727e-10 | -1.671 | 0.170 |

$\ddagger$ p $< 0.01$; $\dagger$ p $< 0.05$; * p $< 0.1$

Table 11: Comparisons of statistical tests

