# OpenReview forum: "Truthfulness in LLMs: A Layer-wise Comparative Analysis of Representation Engineering and Contrast-Consistent Search"
_ICLR.cc/2025/Workshop/BuildingTrust — Submitted to BuildingTrust_

### Official Review · Reviewer_Povu · 2025-02-21
**Layer-wise analysis of Truthfulness directions in LLMs**

**Rating:** 4
**Confidence:** 5

**Review:**

The authors perform layer-wise comparative analysis of truthfulness directions in LLMs using two probing methodologies: RepE and CCS. While the results are interesting, the paper does not address important prior works that have identified and analyzed the truthfulness direction of LLMs across different layers:

(1) Marks, S., & Tegmark, M. (2023). The geometry of truth: Emergent linear structure in large language model representations of true/false datasets.

(2) Liu, Junteng, et al. (2024). "On the universal truthfulness hyperplane inside llms."

While comparing different probing methods to find the truthfulness direction is interesting, I believe that the presented work overlaps significantly with existing literature.

---

### Official Review · Reviewer_ZRrb · 2025-03-02

**Rating:** 4
**Confidence:** 3

**Review:**

This paper applies existing knowledge probing methods, RepE and CCS, to analyze truthfulness representations across layers in Gemma models using five datasets. The experiments examine token position effects, layer-wise truthfulness encoding, and cross-dataset transferability.

Strengths:
* It provides a systematic comparison between the two methods.
* It offers useful insights into how truthfulness representations develop across model layers.

Weaknesses:
* The paper lacks novelty as it primarily applies existing techniques rather than introducing new methods.
* The authors do not adequately position their findings within the broader literature on LLM truthfulness.
* The methodology would be stronger with baseline comparisons.
* This work resembles a course project rather than advancing the research frontier. Additional experiments, novel methods, and clearer implications for LLM development would enhance its contribution to the field.

---

### Decision · Program_Chairs · 2025-03-04

Reject